# Social and environmental determinants of Neglected infectious diseases in quilombola communities of the Brazilian Amazon: An epidemiological and machine learning analysis

Ellen Mara Fernandes da Silva[1], Leanna Silva Aquino[1], Ednaldo Pereira Maranhão[1], Sheyla Mara Silva de Oliveira[1], Tatiane Costa Quaresma[1], Daliane Ferreira Marinho[1], Valney Mara Gomes Conde[1], Veridiana Barreto do Nascimento[1], Irinéia de Oliveira Bacelar Simplício[1], Nádia Vicência do Nascimento Martins[1], Manoel Honorato[1], Adjanny Estela Santos de Souza[1], Franciane de Paula Fernandes[1], Edna Ferreira Coelho Galvão[1], Lívia de Aguiar Valentim[2,3]*

**1** State University of Pará – UEPA, Santarém, Brasil, **2** University of São Paulo (USP), Sao Paulo, Brasil, **3** University of the State of Pará, Santarém, Brasil

☉ These authors contributed equally to this work.
* livia.valentim@uepa.br

## Abstract

Neglected infectious diseases (NIDs) remain a major public health challenge in the Amazon, particularly among quilombola populations living in rural and riverside territories marked by historical inequalities and structural limitations. This study examined the occurrence of NIDs in eight quilombola communities in the Lower Amazon, identified socioenvironmental factors associated with these conditions, and evaluated the performance of machine learning models in predicting individual risk of illness. This analytical cross-sectional study included 518 participants, with data collected through a structured questionnaire. Descriptive and bivariate analyses were conducted, followed by multivariable logistic regression, Poisson regression, cluster analysis, and predictive modeling using Random Forest, XGBoost, and Logistic Regression. Spatial analysis was performed in Google Colab. The overall prevalence of at least one NID was 34.7%. Lack of sanitation facilities, use of river or well water, precarious housing, inadequate waste disposal, low income, and residence in rural areas were significantly associated with both the occurrence and number of NIDs per individual. XGBoost and Random Forest achieved the best predictive performance (AUC-ROC 0.87 and 0.85, respectively). Cluster analysis revealed distinct vulnerability profiles, with the highest burden observed among groups characterized by multidimensional poverty and limited sanitation. The findings highlight the overlapping social and environmental determinants that sustain the persistence of NIDs in these territories, underscoring the need for structural, territorialized policies tailored to the specific realities of quilombola communities in the Amazon. The cross-sectional design and

**Data availability statement:** The data analyzed in this study are not publicly available due to privacy and ethical restrictions. The dataset used for this research contains sensitive patient information and is subject to confidentiality agreements. Therefore, the data cannot be shared openly. Anonymized data and metadata may be made available under controlled access and approval by the ethics committee, e-mail: ceptapajos@uepa.br.

**Funding:** The author(s) received no specific funding for this work.

**Competing interests:** The authors have declared that no competing interests exist.

reliance on self-reported disease history should be considered when interpreting the findings.

## Author summary

Neglected infectious diseases (NIDs) continue to affect populations living in conditions of poverty, especially in remote regions such as the Brazilian Amazon. This study focuses on quilombola communities, Afro-descendant populations that have historically faced social exclusion and limited access to basic services. We investigated how living conditions, such as access to clean water, sanitation, housing quality, and income, are related to the occurrence of NIDs in eight quilombola communities in the Lower Amazon region. Among the 518 adults included in the study, more than one-third reported having had at least one NID during their lifetime. Diseases such as dengue, malaria, leprosy, and parasitic infections were the most frequently reported. We found that people living without a bathroom, using river or well water, residing in precarious housing, and having low income were much more likely to experience these diseases. In addition to traditional epidemiological analyses, we used machine learning models to identify patterns of vulnerability and predict disease risk. These models performed well and highlighted that social and environmental conditions act together to increase health risks. Our findings show that NIDs in quilombola communities are not only biological problems, but also reflect longstanding social and territorial inequalities. Addressing these diseases requires investments in sanitation, water access, housing, and health services tailored to the realities of Amazonian communities.

## Introduction

Neglected infectious diseases (NIDs), understood as infectious conditions strongly associated with poverty, environmental exposure, and historical underprioritization in health systems, remain one of the greatest challenges for health systems in low- and middle-income countries, particularly in regions where state presence is limited and structural inequalities shape everyday life [1,2]. In Brazil, these diseases persist in settings marked by historical vulnerability, unfavorable environmental conditions, and restricted access to essential services [3,4]. In the Amazon, where the territory is vast, mobility depends on the rhythm of the rivers, and many communities live far from urban centers, NIDs find fertile ground to persist, reinforcing cycles of illness that extend across generations [5,6].

Among the most affected groups are quilombola communities, whose historical trajectory carries marks of resistance, but also of political and sanitary invisibility. Living in rural and riverside contexts, these communities face precarious sanitation systems, housing susceptible to vector exposure, unreliable water sources, and daily

difficulties in reaching health services [7]. Geographic distance, lack of regular transportation, low educational levels, and dependence on extractive or agricultural activities increase risks and reduce opportunities for prevention and care [8]. Thus, vulnerability in these territories is not only biological or environmental; it is also social, historical, and spatial.

Despite the relevance of this scenario, few studies have examined in an integrated manner how socioenvironmental, economic, and territorial factors interact in the occurrence of NIDs among Amazonian quilombola populations [7,9]. In many cases, the reality of these communities appears only as footnotes in national statistics, lacking the depth necessary to guide policies that are sensitive to their specificities [10,11]. Understanding health conditions in these territories requires recognizing the role of inequalities, identifying points of greater fragility, and understanding how environment and poverty shape disease exposure [12].

In this context, analyses that combine traditional epidemiological methods with machine learning techniques may offer new perspectives on patterns of illness. These models allow the exploration of complex relationships among social determinants of health, housing conditions, water sources, sanitation, income, and environmental exposures, variables that, in isolation, do not fully explain the persistence of NIDs but, when examined together, reveal deeper layers of vulnerability [13–15].

Given this reality, the present study aimed to analyze the occurrence of NIDs in quilombola communities of the Lower Amazon, identify socioenvironmental factors associated with these conditions, and evaluate to individual risk of illness. We hypothesized that the occurrence of NIDs in these communities is significantly associated with adverse socioenvironmental conditions, particularly inadequate sanitation, unsafe water sources, precarious housing, and low income and that the combined analysis of these determinants through machine learning approaches would allow the identification of distinct vulnerability profiles and improve the prediction of disease risk.

By illuminating aspects of daily life that often remain invisible in public health surveillance, this study seeks to contribute to strengthening disease monitoring, reducing inequities, and informing actions that are more closely aligned with the needs of populations historically marginalized from public policy agendas.

## Methods

### Ethics statement

The study was approved, under approval number 4.915.684 and CAAE 46491621.1.0000.5168, by the Research Ethics Committee (CEP) of the State University of Pará, located at Rua Plácido de Castro, 1399, Bairro Aparecida, Santarém, PA, CEP: 68038610, e-mail: ceptapajos@uepa.br informed consent was obtained for written from all participants and all methods were performed in accordance with current guidelines and standards. Data collection began on April 10, 2024, and continued through October 2025. Written informed consent was obtained from all participants prior to data collection.

This study was grounded in the Theory of Health Inequities, which recognizes that social, economic, and territorial inequalities shape health processes, particularly in historically marginalized populations. This perspective, extensively discussed by Michael Marmot [16], guided the understanding that material conditions, access to essential services, and the social structure of territories decisively influence the distribution of NIDs. Among Amazonian quilombola communities, this approach is particularly relevant, as these groups face structural limitations shaped by ethnic–racial inequalities and persistent geographic barriers.

In this study, the definition of Neglected infectious diseases (NIDs) followed a broad public health perspective, considering conditions that predominantly affect socially vulnerable populations, are strongly associated with poverty and environmental exposure, and remain underprioritized in health systems. In addition to WHO-listed NTDs, diseases such as malaria, viral hepatitis, and parasitic infections were included due to their endemicity in the Amazon region and their persistent association with social and environmental neglect. To address the heterogeneity in transmission routes and morbidity profiles, analyses were conducted at two levels: (i) the occurrence of at least one NID (Any NIDs), capturing

cumulative vulnerability, and (ii) disease-specific analyses for selected conditions, allowing a more granular interpretation of determinants.

We conducted an analytical cross-sectional study with a quantitative approach in eight quilombola communities located in rural and riverside areas of Santarém, in the Lower Amazon region. The selection of communities was designed to represent both upland and floodplain territories, reflecting the municipality's geographic diversity. Data collection took place from across different community settings, including the quilombola primary health unit, local association meetings, and itinerant health activities conducted in partnership with local initiatives. Fieldwork was supported by quilombola high school students funded through scientific initiation programs, which strengthened community engagement and facilitated dialogue with families.

Eligible participants were residents aged 18 years or older who self-identified as quilombolas and lived permanently in the selected communities. Individuals with a minimum of six months of continuous residence in the community were considered permanent residents, a criterion adopted to exclude visitors and temporary residents. Visitors, temporary residents, minors, and individuals unable to complete the questionnaire were excluded.

The sample size was determined based on a sample size calculation using data from the total population of quilombola individuals in the municipality (N = 4,363), of whom 2,133 reside in officially recognized territories. The calculation adopted a 95% confidence level and a 5% margin of error, applying the conventional formula for finite populations $n = N/(1 + N \cdot E^2)$, where N represents the population size and E the margin of error. The total quilombola population of the municipality was chosen as a reference, since the study had a broad territorial scope and sought to capture common structural patterns among recognized communities. This procedure indicated a minimum required sample of approximately 326 participants. Although the sample size calculation did not previously incorporate the community clustering effect, this structure was considered in the analytical stage through the use of robust standard errors adjusted for community clustering.

Participants were recruited through active outreach in the selected quilombola communities, including household visits, community meetings, and activities conducted at the local quilombola primary health unit. All eligible adults present at the time of fieldwork were invited to participate. Of those invited, eight individuals declined participation. As a result, the study included 518 participants, exceeding the estimated sample size and strengthening the representativeness and robustness of the findings.

Data were collected using a structured, validated questionnaire covering sociodemographic information, housing conditions, community infrastructure, environmental factors, and self-reported history of NIDs. The occurrence of Neglected infectious diseases was assessed based on self-reported lifetime history, in which participants reported previous diagnoses made by health professionals. Although this approach allows the capture of cumulative exposure to NIDs in contexts of long-term structural vulnerability, it may introduce recall bias, as past infections may not fully reflect current living conditions. This limitation was acknowledged and addressed by focusing the analyses on persistent socioenvironmental characteristics, such as housing type, water source, sanitation, and waste management, which tend to remain stable over time in these territories.

Data analysis was performed using Stata 16.0 and Python (scikit-learn). Descriptive analyses were conducted to characterize the sample, followed by bivariate analyses conducted to estimate crude associations between socioenvironmental factors and NID occurrence, with effect measures calculated directly from observed frequencies. Multivariable logistic regression and Poisson regression models were used to examine factors associated with the occurrence and number of Neglected infectious diseases. Poisson regression was used because it is a count outcome (number of diseases per individual), allowing the estimation of rate ratios associated with the cumulative burden of NIDs, an approach widely used in epidemiological analyses of this type. All models were adjusted a priori for age and sex, in addition to the main socioenvironmental variables of interest. Given that participants were nested within eight quilombola communities, robust standard errors clustered at the community level were applied to account for intra-cluster correlation and to ensure valid statistical inference. A significance level of 5% was adopted in all models.

The study also included a cluster analysis using unsupervised machine learning techniques to identify vulnerability profiles within the communities based on demographic, socioeconomic, environmental, and housing-related variables, including age, sex, water source, sanitation, housing type, waste disposal, income, area of residence, and presence of domestic animals. This exploratory clustering approach aimed to identify patterns of multidimensional vulnerability without predefined outcome labels. In a subsequent analytical step, supervised machine learning models were applied to predict the occurrence of NIDs. Model training included data standardization, handling of class imbalance through class weighting, and stratified 5-fold cross-validation to ensure robust performance estimates. Predictive performance was evaluated using AUC-ROC, precision, recall, and F1-score.

Spatial analysis was conducted using Google Colab, employing Python geoprocessing libraries to generate a relative risk map integrating NID prevalence and key environmental characteristics. The use of Colab enabled flexible, high-performance computation and reproducible visualization.

The study complied with the ethical guidelines of Resolution No. 466/2012. It was approved by the Research Ethics Committee of the Universidade do Estado do Pará (CAAE No. 46491621.1.0000.5168; Approval No. 4.915.684). All participants were informed about the study objectives and provided written consent. Confidentiality, anonymity, and respect for the cultural specificities of quilombola communities were ensured throughout the research process. Data collection began on April 10, 2024, and continued through October 2025.

## Results

The sociodemographic and sanitation characteristics of the 518 study participants are presented in Table 1. Most participants were women (53.3%), while men represented 46.7% of the sample. A predominance of families with monthly income up to R$ 500.00 was observed (42.1%), followed by those earning between R$ 501 and R$ 1.500 (36.1%). More than half of the households were built with masonry (55.4%), while 33.4% were made of wood or stilt structures and

**Table 1.  Sociodemographic and sanitary characteristics of the studied population in the lower Amazonas communities (n = 518).**

| Variable | Category | n | % |
|---|---|---|---|
| Sex | Male | 242 | 46.7% |
| | Female | 276 | 53.3% |
| Family Income | ≤ R$ 500 | 218 | 42.1% |
| | R$ 501-1,500 | 187 | 36.1% |
| | > R$ 1,500 | 113 | 21.8% |
| Type of Housing | Masonry | 287 | 55.4% |
| | Wood/Stilt House | 173 | 33.4% |
| | Thatch/Adobe | 58 | 11.2% |
| Water Source | Mineral/Treated | 321 | 62.0% |
| | Well | 108 | 20.8% |
| | River | 89 | 17.2% |
| Sanitation | With Bathroom | 476 | 91.9% |
| | Without Bathroom | 42 | 8.1% |
| Waste Disposal | Regular Collection | 391 | 75.5% |
| | Open-air/Burning | 127 | 24.5% |

Legend: This table presents the distribution of participants according to sex, family income, housing characteristics, water source, sanitation facilities, and waste disposal practices in the eight quilombola communities included in the study.

11.2% of straw or clay. Regarding water access, 62.0% of participants reported using mineral or treated water, 20.8% relied on well water, and 17.2% used water directly from the river.

Most households had a bathroom (91.9%), whereas 8.1% lacked sanitary facilities. Regarding waste disposal, 75.5% reported regular waste collection, while 24.5% used open dumping or burning.

The prevalence of NIDs identified in the study is presented in Table 2. Among the investigated diseases, dengue was the most frequent (10.2%), followed by malaria (7.7%), leprosy (5.2%), and parasitic infections (5.0%). Tuberculosis was reported by 4.6% of participants, hepatitis by 3.7%, leishmaniasis by 1.4%, and Chagas disease by 0.6%. Considering the occurrence of at least one NID during their lifetime, 34.7% of the evaluated population had a history of one of these conditions.

The multiple logistic regression analysis (Table 3) identified independent variables associated with the occurrence of any NID. The absence of a bathroom in the household showed the strongest association (OR 5.87; 95%CI: 3.42–10.08).

**Table 2. Prevalence of neglected infectious diseases in the studied Quilombola population (n = 518).**

| Disease | n | % |
|---|---|---|
| Dengue | 53 | 10.2% |
| Malaria | 40 | 7.7% |
| Leprosy | 27 | 5.2% |
| Parasitic Infections | 26 | 5.0% |
| Tuberculosis | 24 | 4.6% |
| Hepatitis | 19 | 3.7% |
| Chagas Disease | 3 | 0.6% |
| Leishmaniasis | 7 | 1.4% |
| Any NID (≥1 disease) | 180 | 34.7% |

Legend: This table shows the lifetime self-reported prevalence of individual Neglected infectious diseases and the proportion of participants reporting at least one NID.

**Table 3. Multiple logistic regression analysis for factors associated with the occurrence of any NID (n = 518).**

| Variable | Category | OR | 95% CI | p-value |
|---|---|---|---|---|
| Sanitation | Without bathroom | 5.87 | 3.42–10.08 | <0.001 |
| | With bathroom (ref.) | 1.00 | — | — |
| Water source | River | 4.32 | 2.15–8.67 | <0.001 |
| | Well | 2.14 | 1.08–4.24 | 0.029 |
| | Treated water (ref.) | 1.00 | — | — |
| Type of housing | Non-masonry | 4.12 | 2.54–6.68 | <0.001 |
| | Masonry (ref.) | 1.00 | — | — |
| Waste disposal | Inadequate | 3.45 | 2.18–5.46 | <0.001 |
| | Regular collection (ref.) | 1.00 | — | — |
| Area of residence | Rural | 2.89 | 1.82–4.59 | <0.001 |
| | Urban (ref.) | 1.00 | — | — |
| Family income | ≤ R$ 1,500 | 2.56 | 1.61–4.07 | <0.001 |
| | > R$ 1,500 (ref.) | 1.00 | — | — |
| Sex | Female | 0.89 | 0.62–1.28 | 0.527 |
| | Male (ref.) | 1.00 | — | — |

Legend: Odds ratios (OR) and 95% confidence intervals (CI) were estimated using multivariable logistic regression. Reference categories are indicated as (ref.). The model evaluates independent associations between socioenvironmental determinants and the presence of at least one NID.

Subsequently, the use of river water (OR 4.32; 95%CI: 2.15–8.67) and well water (OR 2.14; 95%CI: 1.08–4.24) were also relevant factors. Poor housing conditions, characterized by dwellings not built with masonry, showed an OR of 4.12 (95%CI: 2.54–6.68). Inadequate waste disposal increased the risk of NIDs (OR 3.45; 95%CI: 2.18–5.46). Residing in a rural area (OR 2.89) and having a monthly family income of up to R$ 1,500.00 (OR 2.56) were also associated with the occurrence of NIDs. The variable sex showed no statistically significant association.

The Poisson regression presented in Table 4 showed that the absence of a bathroom (IRR 1.92; 95%CI: 1.65–2.24) was associated with an increase in the number of NIDs per individual. The use of river water (IRR 1.86) and well water (IRR 1.32) also showed a significant relationship. Other factors that increased the number of diseases included: dwellings not built with masonry (IRR 1.74), lack of regular waste collection (IRR 1.58), residence in a rural area (IRR 1.45), and a family income less than R$ 1,500.00 (IRR 1.42). As in the logistic regression, sex showed no significant association.

The performance of machine learning algorithms in predicting NIDs is presented in Table 5. The aggregate outcome 'Any NID' represents a synthetic indicator of cumulative vulnerability and does not replace the disease-specific analyses, which are presented in a complementary manner. For the outcome "any NID," the XGBoost model showed the best performance, with an AUC-ROC of 0.87, followed by Random Forest (AUC-ROC of 0.85). The Logistic Regression model had an AUC-ROC of 0.79. For specific outcomes, such as dengue and hepatitis, Random Forest performed well, with AUC-ROC ranging between 0.80 and 0.82. These findings indicate relevant predictive capacity of the models, especially those based on ensemble techniques.

The bivariate analysis presented in Table 6 showed substantial differences in the proportion of NIDs between the evaluated categories. The absence of a bathroom showed the highest proportion of cases (90.5%), followed by the use of river water (50.6%). Inadequate waste disposal and poor housing conditions also showed higher proportions of diseases when compared to the reference categories. These analyses complement the multivariate findings, showing consistent patterns among the main socio-environmental factors. The categories used in the bivariate analysis were maintained at a

**Table 4. Poisson regression analysis for factors associated with the number of NIDs per individual (n = 518).**

| Variable | Category | IRR | 95% CI | p-value |
|---|---|---|---|---|
| Sanitation | Without bathroom | 1.92 | 1.65–2.24 | <0.001 |
| | With bathroom (ref.) | 1.00 | — | — |
| Water source | River | 1.86 | 1.52–2.28 | <0.001 |
| | Well | 1.32 | 1.06–1.64 | 0.013 |
| | Treated water (ref.) | 1.00 | — | — |
| Type of housing | Non-masonry | 1.74 | 1.51–2.01 | <0.001 |
| | Masonry (ref.) | 1.00 | — | — |
| Waste disposal | Inadequate | 1.58 | 1.38–1.81 | <0.001 |
| | Regular collection (ref.) | 1.00 | — | — |
| Area of residence | Rural | 1.45 | 1.26–1.67 | <0.001 |
| | Urban (ref.) | 1.00 | — | — |
| Family income | ≤ R$ 1,500 | 1.42 | 1.23–1.64 | <0.001 |
| | > R$ 1,500 (ref.) | 1.00 | — | — |
| Sex | Female | 0.96 | 0.85–1.08 | 0.478 |
| | Male (ref.) | 1.00 | — | — |
| | | | | |

Legend: Incidence rate ratios (IRR) and 95% confidence intervals (CI) were estimated using Poisson regression. Reference categories are indicated as (ref.). This model assesses the association between socioenvironmental factors and the cumulative number of NIDs reported by each participant.

**Table 5. Performance of machine learning algorithms in predicting NIDs (n = 518).**

| Model | Disease/Target | AUC-ROC | Precision | Recall | F1-Score |
|---|---|---|---|---|---|
| Random Forest | Any NID | 0.85 | 0.56 | 0.60 | 0.58 |
| Logistic Regression | Any NID | 0.79 | 0.48 | 0.50 | 0.49 |
| XGBoost | Any NID | 0.87 | 0.59 | 0.63 | 0.61 |
| Random Forest | Dengue | 0.82 | 0.49 | 0.53 | 0.51 |
| Random Forest | Hepatitis | 0.80 | 0.44 | 0.48 | 0.46 |

Legend: Model performance was assessed using the area under the receiver operating characteristic curve (AUC-ROC), precision, recall, and F1-score. Ensemble-based models were compared with logistic regression for predicting NID occurrence.

**Table 6. Crude associationbetween socio-environmental factors and occurrence of NIDs (n = 518).**

| Risk Factor | Category | With NID n/N | % | Crude OR (95% CI) |
|---|---|---|---|---|
| Bathroom | Without | 38/42 | 90.5% | 22.35 (7.83–63.78) |
| | With | 142/476 | 29.8% | 1.00 (ref.) |
| Water Source | River | 45/89 | 50.6% | 2.05 (1.27–3.29) |
| | Well | 28/108 | 25.9% | 0.70 (0.43–1.14) |
| | Treated | 107/321 | 33.3% | 1.00 (ref.) |
| Waste Destination | Open-air/Burning | 52/127 | 40.9% | 1.42 (0.94–2.15) |
| | Collected | 128/391 | 32.7% | 1.00 (ref.) |
| Type of Housing | Precarious | 58/173 | 33.5% | 0.92 (0.63–1.36) |
| | Adequate | 122/345 | 35.4% | 1.00 (ref.) |
| Income | ≤ R$ 500 | 61/218 | 28.0% | 0.59 (0.41–0.86) |
| | > R$ 500 | 119/300 | 39.7% | 1.00 (ref.) |

Legend: Crude odds ratios (OR) and 95% confidence intervals (CI) were calculated based on observed frequencies. Reference categories are indicated as (ref.).

higher level of disaggregation for exploratory purposes, whereas the multivariable models adopted aggregated categories to ensure statistical stability.

The cluster segmentation, presented in Table 7, identified four distinct groups. The multidimensional poverty cluster showed the highest prevalence of NIDs (52.3%), followed by the sanitary vulnerability cluster (41.7%). The vulnerable elderly cluster had a prevalence of 38.9%, while the low-risk cluster recorded a prevalence of 8.2%.

The cross-validation results (Table 8) show that Random Forest achieved the best overall performance for most of the analyzed diseases, particularly for malaria (AUC 0.81), tuberculosis (AUC 0.76), and parasitic infections (AUC 0.73). Logistic Regression showed inferior performance across all outcomes. Although the category "Any NID" aggregates diseases with distinct transmission mechanisms, it was used as a synthetic indicator of cumulative exposure to neglected health conditions. Disease-specific analyses were therefore presented to account for etiological heterogeneity.

The variable importance analysis (Table 9) revealed that the most relevant factors for the Random Forest model were: water supply (14.3%), type of housing (11.7%), area of residence (10.5%), and sanitation (9.8%). Other important factors included income, waste collection, and the presence of domestic animals. The occupation variable was not included in the initial sociodemographic characterization due to its high heterogeneity and informality, but it was incorporated into the predictive models because of its discriminatory potential in the context of territorial vulnerability.

Fig 1 presents the map of Santarém, with a territorial area of 17,898.389 km², showing the spatial distribution of the relative risk for NIDs in the analyzed quilombola communities. It is observed that localities such as Tiningu and Arapemã

**Table 7. Population segmentation by vulnerability clusters (n = 518).**

| Cluster | n | % | Characteristics | NID Prevalence |
|---|---|---|---|---|
| Low Risk | 166 | 32.0% | Adequate sanitation, medium/high income | 8.2% |
| Sanitary Vulnerability | 145 | 28.0% | Untreated water, precarious sanitation | 41.7% |
| Multidimensional Poverty | 130 | 25.1% | Low income, precarious housing, low education | 52.3% |
| Vulnerable Elderly | 77 | 14.9% | Age ≥ 60 years, comorbidities | 38.9% |

Legend: Clusters were derived using unsupervised machine learning techniques to identify groups with similar socioenvironmental characteristics. No reference category applies. NID prevalence is presented for descriptive comparison across clusters.

**Table 8. Average model performance metrics (5-fold cross-validation).**

| Disease/ Target | Model | AUC ROC | Avg. Precision (AP) | F1 | Precision | Recall |
|---|---|---|---|---|---|---|
| Malaria | Random Forest | 0.81 | 0.62 | 0.56 | 0.58 | 0.54 |
| Malaria | Logistic Regression | 0.78 | 0.59 | 0.49 | 0.51 | 0.48 |
| Tuberculosis | Random Forest | 0.76 | 0.57 | 0.44 | 0.46 | 0.43 |
| Chagas Disease | Random Forest | 0.74 | 0.54 | 0.41 | 0.44 | 0.40 |
| Leishmaniasis | Random Forest | 0.71 | 0.50 | 0.39 | 0.41 | 0.38 |
| Leprosy | Random Forest | 0.72 | 0.52 | 0.40 | 0.43 | 0.39 |
| Parasitic Infections | Random Forest | 0.73 | 0.55 | 0.42 | 0.44 | 0.40 |
| Any NID | Random Forest | 0.83 | 0.69 | 0.58 | 0.60 | 0.57 |
| Any NID | Logistic Regression | 0.79 | 0.64 | 0.51 | 0.53 | 0.50 |

Legend: Model performance metrics are presented as averages across stratified 5-fold cross-validation. AUC-ROC, average precision (AP), F1-score, precision, and recall were used to evaluate predictive accuracy for different NID outcomes.

**Table 9. Most influential factors associated with the occurrence of neglected diseases (Random Forest).**

| Variable/ Factor | Relative Importance (%) |
|---|---|
| Water Supply | 14.3 |
| Type of Housing | 11.7 |
| Area of Residence (Rural) | 10.5 |
| Sanitation | 9.8 |
| Waste Collection | 8.6 |
| Family Income | 7.9 |
| Presence of Domestic Animals | 7.4 |
| Age | 6.8 |
| Education Level | 6.1 |
| Occupation (Rural or Informal) | 5.9 |

Legend: Variable importance values represent the relative contribution of each predictor to the Random Forest model. Higher percentages indicate greater influence on model predictions.

show the highest risk levels, classified in the critical category, while communities such as Pérola do Maicá and Saracura are situated in the lower risk categories. The figure integrates information on the proportion of NIDs and the estimated risk, graphically represented by variations in color and marker size, allowing for the visualization of territorial heterogeneity among the communities.

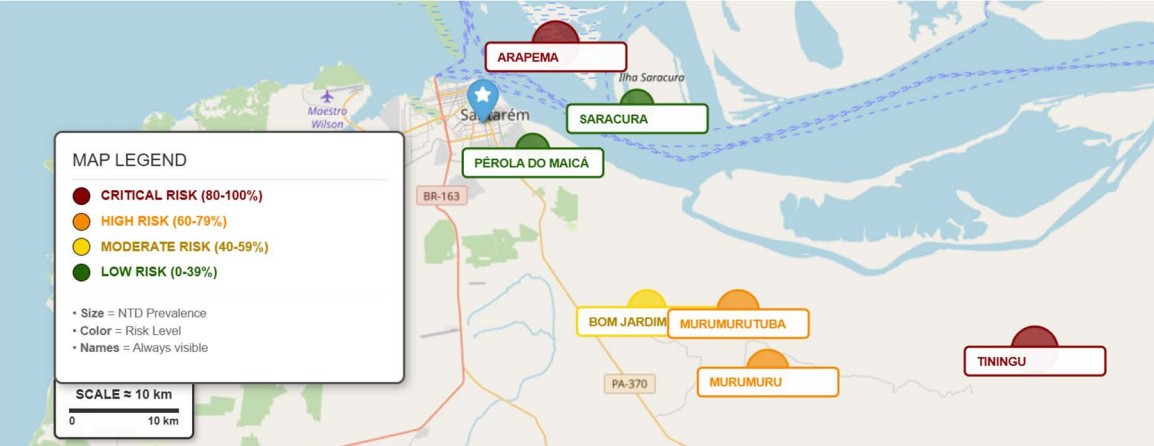

**Fig 1. Map of the spatial distribution of neglected infectious disease risk in the quilombola communities of the lower Amazonas.** The map illustrates the territorial distribution of relative risk for neglected infectious diseases (NIDs) across eight quilombola communities in the municipality of Santarém (17,898.389 km²). Risk levels are represented through color gradients and proportional markers, integrating disease prevalence and estimated relative risk. Communities such as Tiningu and Arapemã are classified in the highest risk category, while Pérola do Maicá and Saracura present lower relative risk levels. The figure highlights territorial heterogeneity and spatial inequalities in disease distribution within the Lower Amazon region. Source: https://www.ibge.gov.br/geociencias/organizacao-do-territorio/estrutura-territorial/15761-localidades.html.

## Discussion

The findings of this study indicate that neglected infectious diseases (NIDs) remain deeply intertwined with the structural conditions that shape daily life in quilombola communities of the Lower Amazon. The high prevalence observed and the strong associations with inadequate sanitation, unsafe water sources, precarious housing, and low income highlight how multiple infectious diseases coexist and accumulate in territorially vulnerable settings. Although only a subset of the conditions analyzed is formally classified as Neglected Tropical Diseases by the World Health Organization, the broader category of NIDs adopted here better reflects the epidemiological reality of the Amazon, where endemic infectious diseases share common social and environmental determinants and disproportionately affect historically marginalized populations.

The prevalence of NIDs observed, affecting more than one-third of the population, is consistent with previous studies reporting the persistence of these conditions in territories marked by poverty, historical exclusion, and limited access to essential services [17]. This pattern has been widely documented in hard-to-reach regions, where physical and social environments intersect to sustain prolonged cycles of infectious disease transmission [18].

The factors associated with illness underscore the predominance of environmental and social determinants over strictly individual characteristics, aligning with the literature that describes NIDs as diseases linked to poverty and environmental precariousness [19,20]. The strong association between lack of sanitation and increased NID risk confirms well-established evidence that inadequate wastewater management facilitates the spread of parasitic and environmentally transmitted diseases [21]. Studies conducted in other Amazonian regions show similar results, highlighting the central role of sanitation as a protective barrier [22].

Water supply emerged as one of the most relevant determinants, particularly for individuals relying on river or well water. This relationship is well documented in the literature, especially in the context of waterborne diseases such as parasitic infections, diarrheal diseases, and viral hepatitis, which remain major public health issues along the Amazon basin [23]. Vulnerability associated with prolonged exposure to untreated water is intensified by logistical challenges and the absence of state infrastructure in these areas, as described in studies of traditional and riverside populations [24].

Housing precariousness identified as a risk factor in this study has likewise been consistently associated with greater exposure to disease vectors. Wooden houses, clay structures, and stilted dwellings increase the likelihood of mosquito, rodent, and triatomine infiltration, facilitating the transmission of dengue, malaria, and Chagas disease, as demonstrated in studies [25,26]. This structural vulnerability is further exacerbated by the difficulty of implementing housing improvements in isolated communities, where transportation and access to building materials pose significant challenges [27].

Improper waste disposal, another factor associated with illness, aligns with patterns reported in research on dengue, leishmaniasis, and other infections that benefit from environments conducive to vector proliferation [28]. The literature indicates that irregular waste collection in rural, quilombola, and riverside territories is directly linked to the absence of continuous public policies and the historical invisibility of these populations within municipal sanitation strategies [29].

Cluster analysis revealed that risk is not evenly distributed within the communities, but stratified according to socioeconomic and environmental conditions. The cluster characterized by multidimensional poverty showed the highest prevalence of NIDs, consistent with studies demonstrating that the overlap of vulnerabilities, low income, low education, and precarious housing, intensifies risk and sustains intergenerational cycles of illness [30]. International literature similarly indicates that Black and traditional populations, particularly those geographically isolated, experience greater exposure and reduced access to timely diagnosis [31,32].

Machine learning models added an innovative perspective to understanding NID determinants. The strong performance of XGBoost and Random Forest aligns with recent studies emphasizing the potential of these techniques to identify complex patterns in contexts of high vulnerability and limited structured data [33]. The convergence between factors identified by the models and those revealed through traditional statistical analyses strengthens the robustness of the findings and reinforces that physical and social environments play decisive roles in NID distribution.

The territorial risk map highlights the spatial inequality that permeates the Lower Amazon. Communities located farther from urban centers, such as Tiningu and Arapemã, exhibited critical risk levels, consistent with research on geographic access showing that long distances, lack of regular transportation, and river seasonality significantly increase barriers to care [34,35]. This territorial configuration results in what the literature refers to as "double vulnerability," in which historically marginalized populations also experience geographic neglect [36].

Taken together, these findings indicate that addressing NIDs in these territories requires more than biomedical interventions. The literature consistently highlights that structural actions, adequate sanitation, access to treated water, environmental management, and housing improvements, play decisive roles in reducing these diseases [1,37]. In the Amazonian quilombola context, such policies must consider the realities of riverside life, territorial geography, and community autonomy, avoiding standardized, urban-centered solutions that have historically failed to reach traditional populations [7,38].

Some analytical limitations should be acknowledged when interpreting the findings. The occurrence of Neglected infectious diseases was assessed based on self-reported lifetime history, which may be subject to recall bias and may not fully reflect current exposure conditions. In addition, the cross-sectional design of the study precludes causal inference, limiting the interpretation of observed relationships to associations. Although the analyses were adjusted for key confounders, including age, sex, and community-level clustering, residual confounding related to unmeasured or imprecisely measured factors cannot be ruled out. Finally, the relatively small number of events for some specific diseases may have reduced statistical power and precision in disease-specific analyses. These limitations should be considered when generalizing the results, although the consistency of findings across multiple analytical approaches supports the robustness of the observed associations.

## Conclusion

The results of this study suggest that the persistence of NIDs in quilombola communities of the Lower Amazon is strongly linked to the structural conditions that shape everyday life, including insufficient sanitation, inadequate water supply, precarious housing, and limitations in waste management. These conditions reflect longstanding inequities and reveal vulnerabilities that may be mitigated through more consistent and territorially sensitive public policies.

The integrated analysis, including machine learning techniques, shows that socioenvironmental determinants do not act in isolation but combine to amplify disease risk. These patterns indicate that isolated interventions, such as seasonal prevention campaigns, are unlikely to be sufficient for NIDs in populations living far from urban centers and facing continuous logistical challenges to accessing care. The findings also show that in contexts where surveillance capacity is limited, advanced analytical tools can help identify priority areas, without replacing the need for long-term structural actions.

The territorial heterogeneity observed across communities illustrates that disease risk is not uniformly distributed. More remote areas, characterized by limited public services and environmental precariousness, concentrated the highest levels of risk. This observation, already reflected in studies on traditional and riverside populations, indicates that geographic distance remains an important marker of health inequity. Progress in reducing NIDs in these contexts requires acknowledging that logistical, geographic, and socioeconomic challenges directly influence the effectiveness of health actions.

Taken together, the findings underscore the need for integrated and sustainable strategies tailored to the socioterritorial specificities of the Amazon. Investments in adequate sanitation, access to treated water, housing improvements, and strengthened primary care are essential to reducing the inequities identified. Furthermore, epidemiological surveillance policies should incorporate territorialized approaches capable of anticipating scenarios and guiding the allocation of resources more effectively. Overcoming these diseases requires not only clinical actions or isolated campaigns, but a sustained agenda addressing the historical inequalities that underpin the burden of NIDs in the Amazon.

From an operational standpoint, these findings can support the territorial prioritization of health surveillance actions, the allocation of resources for sanitation in higher-risk communities, and the strengthening of primary care strategies sensitive to the specific needs of quilombola communities, especially in more remote areas of the municipality.

## Supporting information

**S1 Data. Anonymized dataset containing the variables analyzed in this study, including sociodemographic, environmental, and health-related data, along with the corresponding codebook describing variable definitions, categories, and coding procedures used in the statistical and machine learning analyses.**
(XLSX)

## Acknowledgments

We would like to thank the partnership with the project "Weaving Networks of Health and Knowledge: Innovation and Tradition in the Amazon", approved through a public call by the National Council for Scientific and Technological Development (CNPq) and carried out by the University of State Pará (UEPA).

## Author contributions

**Conceptualization:** Ellen Mara Fernandes da Silva, Leanna Silva Aquino, Ednaldo Pereira Maranhão, Sheyla Mara Silva de Oliveira, Tatiane Costa Quaresma, Daliane Ferreira Marinho, Valney Mara Gomes Conde, Veridiana Barreto do Nascimento, Irinéia de Oliveira Bacelar Simplício, Nádia Vicência do Nascimento Martins, Marcos Manoel Honorato, Adjanny Estela Santos de Souza, Franciane de Paula Fernandes, Edna Ferreira Coelho Galvão, Livia de Aguiar Valentim.

**Formal analysis:** Ellen Mara Fernandes da Silva, Leanna Silva Aquino, Ednaldo Pereira Maranhão, Sheyla Mara Silva de Oliveira, Tatiane Costa Quaresma, Daliane Ferreira Marinho, Valney Mara Gomes Conde, Veridiana Barreto do Nascimento, Irinéia de Oliveira Bacelar Simplício, Nádia Vicência do Nascimento Martins, Marcos Manoel Honorato, Adjanny Estela Santos de Souza, Franciane de Paula Fernandes, Edna Ferreira Coelho Galvão, Livia de Aguiar Valentim.

**Investigation:** Livia de Aguiar Valentim.

**Methodology:** Ellen Mara Fernandes da Silva, Leanna Silva Aquino, Ednaldo Pereira Maranhão, Sheyla Mara Silva de Oliveira, Tatiane Costa Quaresma, Daliane Ferreira Marinho, Valney Mara Gomes Conde, Veridiana Barreto do Nascimento, Irinéia de Oliveira Bacelar Simplício, Nádia Vicência do Nascimento Martins, Marcos Manoel Honorato, Adjanny Estela Santos de Souza, Franciane de Paula Fernandes, Edna Ferreira Coelho Galvão, Livia de Aguiar Valentim.

**Writing – review & editing:** Ellen Mara Fernandes da Silva, Leanna Silva Aquino, Ednaldo Pereira Maranhão, Sheyla Mara Silva de Oliveira, Tatiane Costa Quaresma, Daliane Ferreira Marinho, Valney Mara Gomes Conde, Veridiana Barreto do Nascimento, Irinéia de Oliveira Bacelar Simplício, Nádia Vicência do Nascimento Martins, Marcos Manoel Honorato, Adjanny Estela Santos de Souza, Franciane de Paula Fernandes, Edna Ferreira Coelho Galvão, Livia de Aguiar Valentim.

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
