## [Decision Letter · Decision Letter 0]

30 Dec 2025

Social and environmental determinants of neglected tropical diseases in quilombola communities of the brazilian amazon: an epidemiological and machine learning analysis

Dear Dr. Valentim,

Thank you for submitting your manuscript to PLOS Neglected Tropical Diseases. After careful consideration, we feel that it has merit but does not fully meet PLOS Neglected Tropical Diseases's publication criteria as it currently stands. Therefore, we invite you to submit a revised version of the manuscript that addresses the points raised during the review process.

Please submit your revised manuscript within by Feb 28 2026 11:59PM. If you will need more time than this to complete your revisions, please reply to this message or contact the journal office at plosntds@plos.org. Please include the following items when submitting your revised manuscript:

We look forward to receiving your revised manuscript.

Kind regards,

Abdullahi Yusuf, PhD

Academic Editor

Elsio Wunder Jr

Section Editor

Shaden Kamhawi

co-Editor-in-Chief

Paul Brindley

co-Editor-in-Chief

**Additional Editor Comments (if provided):**

**Journal Requirements:**

**Reviewers' Comments:**

Reviewer's Responses to Questions

**Key Review Criteria Required for Acceptance?**

**Methods**

-Are the objectives of the study clearly articulated with a clear testable hypothesis stated?

-Is the study design appropriate to address the stated objectives?

-Is the population clearly described and appropriate for the hypothesis being tested?

-Is the sample size sufficient to ensure adequate power to address the hypothesis being tested?

-Were correct statistical analysis used to support conclusions?

-Are there concerns about ethical or regulatory requirements being met?

Reviewer #1: The study objectives are clearly stated, and the hypothesis is well aligned with the research aims. The analytical cross-sectional design is appropriate for exploring socioenvironmental determinants of NTDs in quilombola communities. The population is well described, and the sampling strategy is consistent with the study aims. With 518 participants, the sample size is adequate for both the regression models and the machine-learning analyses presented.

The statistical methods are appropriate and correctly applied. The authors use descriptive, bivariate, multivariable logistic regression, Poisson regression, clustering, and machine-learning models in a coherent way. The conclusions are supported by the results.

Ethical approval is clearly reported, including the CAAE number, approval number, and informed consent procedures. There are no concerns regarding ethical or regulatory compliance.

Overall, the methods are sound, and the manuscript meets the criteria for acceptance. Only minor clarifications such as briefly noting how the minimum sample size was calculated may strengthen the methods section but are not essential for acceptance.

Reviewer #2: Although the manuscript tackles an extremely important topic, a few key issues need attention before it can meet PLOS acceptance standards. First, the objectives are clear, but there’s no explicit testable hypothesis, so it would help to add one simple statement about what the study expected to find. The design fits the research question, but because NTDs were measured through self-reported lifetime history, there’s a risk that past infections don’t match current living conditions clarifying how these outcomes were asked and openly acknowledging recall bias will strengthen the paper. The population is well described, but readers still cannot see how participants were recruited or how many refused, so adding a short flow description will make the sample more transparent. The sample size is solid, yet the main analyses need revisiting: important confounders like age were not adjusted for, the data are clustered within eight communities, and some bivariate numbers (like odds ratios in Table 6) don’t match the raw counts re-running the models with age, sex, and cluster-robust errors, and correcting the table values will greatly improve credibility. The machine-learning section is interesting but under-explained, so briefly outlining the variables used, how cross-validation was done, and how class imbalance was handled will make the results more trustworthy. Finally, the ethics and data-sharing pieces need harmonizing: the manuscript reports written consent, yet the “Informed Consent” field says “Not applicable,” and the data-collection dates contradict each other; also, the current data-availability statement doesn’t align with PLOS requirements, so providing a de-identified public dataset or a compliant access process is essential. Addressing these points will make the manuscript much stronger, clearer, and fully aligned with journal expectations.

Reviewer #3: No comment

**Results**

-Does the analysis presented match the analysis plan?

-Are the results clearly and completely presented?

-Are the figures (Tables, Images) of sufficient quality for clarity?

Reviewer #1: The analyses conducted are consistent with the planned methods, and the statistical approach matches what is described in the Methods section. The results are clearly presented, with complete tables and appropriate effect measures. All tables and the spatial figure are of adequate quality and should typeset well, with only minor formatting improvements expected during production.

Reviewer #2: The Results section is rich and informative, but a few issues need tightening to fully match the analysis plan and improve clarity. While the authors do present logistic regression, Poisson models, clustering, and machine-learning results as promised in the Methods, some outputs don’t fully align for example, the clustering method isn’t described earlier, and the ML steps lack detail, so the Results feel slightly disconnected from the stated plan. Several tables are clear, but Table 6 has numerical inconsistencies (crude ORs don’t match raw proportions), which should be corrected to maintain trust in the findings. The regression tables also need clearer labeling of reference categories so readers can easily interpret directionality. Overall, the results are complete, but adding brief explanations of what each model tells us and cleaning up formatting would make them more reader-friendly. Figures are generally acceptable, though the spatial risk map needs higher resolution and a clearer legend. With these adjustments, the Results section would become much more coherent, transparent, and aligned with the analytic strategy.

Reviewer #3: I noted that diseases like Malaria, Hepatitis, and parasitic disease without precision have been consider as NTDs. Those disease must be excluded from the list of NTD identified in the study.

A significant number of "Any NTD" representing 34% must be detailed for a better understanding of NTDs prevalent in the area and the determining factors since transmission, morbidity associated are different.

Those clarification will help avoid biais in the analysis of results

**Conclusions**

-Are the conclusions supported by the data presented?

-Are the limitations of analysis clearly described?

-Do the authors discuss how these data can be helpful to advance our understanding of the topic under study?

-Is public health relevance addressed?

Reviewer #1: The conclusions are well supported by the data and follow logically from the analyses. The authors clearly describe the study’s limitations and acknowledge the constraints of cross-sectional data and self-reported outcomes. The discussion effectively links the findings to broader evidence and shows how the results advance understanding of socioenvironmental determinants of NTDs. Public health relevance is clearly addressed, emphasizing structural vulnerabilities and the need for targeted policies.

Reviewer #2: The conclusions are thoughtful and generally in line with the study’s findings, but a few refinements would make them more solidly grounded in the data. The authors appropriately highlight sanitation, water access, housing, and poverty as major drivers of NTD vulnerability, but they sometimes overstate the strength of causal inferences given the cross-sectional design softening the language and tying claims more strictly to associations would improve accuracy. The limitations section touches on structural inequities, but it does not clearly acknowledge key analytical limitations, such as self-reported lifetime NTDs, recall bias, missing confounders, lack of clustering adjustment, and the small number of events for some diseases. Expanding this section briefly would make the conclusions more balanced. On the positive side, the discussion does a nice job connecting the results to broader themes of health equity and territorial vulnerability, and it clearly addresses public-health relevance by calling for structural and context-sensitive interventions. Strengthening the limitations and ensuring claims stay aligned with the actual analyses will make the conclusions more credible while keeping their meaningful insights intact.

Reviewer #3: No comment

**Editorial and Data Presentation Modifications?**

Reviewer #1: The manuscript is clear and well organized. Only minor editorial adjustments are needed, such as small formatting improvements and brief clarification of the sample size basis. These are not essential for interpretation. I recommend acceptance.

Reviewer #2: Overall, the manuscript is well written and engages with an important public-health topic, but several editorial and presentation improvements would greatly enhance clarity. The most pressing fixes involve correcting the numerical inconsistencies in Table 6, clarifying reference categories in the regression tables, and improving the resolution and legend of the spatial figure. A light language edit to shorten long sentences and tighten flow especially in the Introduction and Discussion would also help readability. The Methods section would benefit from clearer organization, separating epidemiologic, clustering, and machine-learning procedures so the analysis plan aligns cleanly with the Results. Additionally, the contradictory data-collection dates and the incorrect “Informed Consent: Not applicable” entry need correction for transparency. While these issues are largely editorial, they touch enough parts of the manuscript (tables, figures, Methods, ethics statements) that a Major Revision is the most appropriate recommendation to ensure accuracy and coherence before publication.

Reviewer #3: Table are well presented. Need to take into account suggestion made

**Summary and General Comments**

Reviewer #1: This is a strong and well-executed study with clear public health relevance. The integration of traditional epidemiological analyses with machine-learning models adds value and provides novel insights into socioenvironmental determinants of NTDs in quilombola communities. The manuscript is well structured, the methods are appropriate, and the conclusions are supported by the data. Ethical approval and procedures are clearly documented.

Reviewer #2: (No Response)

Reviewer #3: The study is of interest but there is a need to considere NTDs defined by WHO to ensure precision of disease prevalence and determining factors

PLOS authors have the option to publish the peer review history of their article (what does this mean? ). If published, this will include your full peer review and any attached files.

**Do you want your identity to be public for this peer review?** For information about this choice, including consent withdrawal, please see our Privacy Policy .

Reviewer #1: **Yes:** Melanie Awino Abongo

Reviewer #2: **Yes:** Dr.Arman Abdous (DVM,MPH)

Reviewer #3: No

**Figure resubmission:**
---

## [Decision Letter · Decision Letter 1]

8 Feb 2026

Response to Reviewers
Revised Manuscript with Track Changes
Manuscript

Shaden Kamhawi

co-Editor-in-Chief

Paul Brindley

co-Editor-in-Chief

**Additional Editor Comments (if provided):**
**Journal Requirements:**

**Reviewers' comments:**

**Key Review Criteria Required for Acceptance?**

**Methods**

-Are the objectives of the study clearly articulated with a clear testable hypothesis stated?

-Is the study design appropriate to address the stated objectives?

-Is the population clearly described and appropriate for the hypothesis being tested?

-Is the sample size sufficient to ensure adequate power to address the hypothesis being tested?

-Were correct statistical analysis used to support conclusions?

-Are there concerns about ethical or regulatory requirements being met?

Reviewer #1: THE METHOD SECTION IS ADEQUATELY DESCRIBED

Reviewer #2: This revised version shows clear effort from the authors to address prior concerns, and the Methods section is noticeably stronger than in the original submission. The study objectives are now clearly articulated and the addition of an explicit, testable hypothesis at the end of the Introduction is a welcome and important improvement. The cross sectional design remains appropriate for the stated aims, and the description of the quilombola population is clear and well aligned with the research questions. The sample size is clearly sufficient in absolute terms and exceeds the minimum calculated requirement, which strengthens confidence in the analyses. That said, the sample size calculation would benefit from a brief clarification regarding which population size was used in the formula and why, as well as a short comment on whether clustering by community was considered at the planning stage. The statistical methods are generally well chosen and clearly described, but a brief justification for the use of Poisson regression for the count outcome would further strengthen methodological transparency. Ethical approval and informed consent are appropriately documented, although the reported data collection end date should be made fully consistent across the Methods and ethics statements to avoid confusion at this stage of revision.

Reviewer #3: 1. The definition of NTDs in this study is different from WHO definition. I suggest a different name is proposed to make it clear the group of disease targeted by the study. I have observed that 16% of diseases targeted in the study relates to NTD WHO list.

That will avoid confusion on the definition of NTDs.

2. Exlusion criteria mentioned "Visitors, temporary residents". Is there a minimum time of staying in the community that enables exclusion ?

**Results**

-Does the analysis presented match the analysis plan?

-Are the results clearly and completely presented?

-Are the figures (Tables, Images) of sufficient quality for clarity?

Reviewer #1: YES THIS IS WELL DESCRIBED

Reviewer #2: The Results section is clearly improved compared with the previous version and follows a logical structure that mirrors the analytical plan outlined in the Methods. Descriptive, bivariate, multivariable, and machine learning analyses are presented in a coherent sequence, making the findings easy to follow. One remaining issue concerns the bivariate analysis table, where the categorization of income and housing differs from that used in the multivariable models. This leads to apparent reversals in the direction of some crude associations, which may confuse readers even though the multivariable results are sound. Aligning category definitions across tables or explicitly explaining the rationale for different groupings would substantially improve clarity. Tables and figures are generally of good quality and readable, and the spatial risk map adds value to the manuscript, though minor refinements in labeling and attribution would strengthen presentation. The machine learning results are clearly summarized, but a brief reminder within the Results of how the aggregated outcome differs conceptually from disease specific outcomes would help guide interpretation.

Reviewer #3: One of the key determinant relates to occupation that was not taken into account in the Table#1 vs Table#9. Is there a reason why "occupation" was not taken into account in the socio demographic characteristics. This variable can impact health.

**Conclusions**

-Are the conclusions supported by the data presented?

-Are the limitations of analysis clearly described?

-Do the authors discuss how these data can be helpful to advance our understanding of the topic under study?

-Is public health relevance addressed?

Reviewer #1: THIS SECTION IS ADEQUATELY ADRESSED

Reviewer #2: The conclusions are appropriately cautious and are supported by the data presented, reflecting a clear improvement in tone compared with the original submission. The authors now avoid causal language and explicitly acknowledge key limitations, including recall bias, the cross sectional design, and potential residual confounding, which strengthens the credibility of the interpretation. The discussion successfully places the findings within the broader literature and highlights how social and environmental determinants shape NTD vulnerability in quilombola communities. Public health relevance is clearly articulated, particularly regarding sanitation, housing, and territorial inequities. The conclusions could be further strengthened by adding one concrete example of how these findings might inform local surveillance, resource allocation, or program prioritization, which would make the implications even more tangible for policy oriented readers.

Reviewer #3: No commment

**Editorial and Data Presentation Modifications?**

Reviewer #1: INCLUDE THE STUDY LIMITATION IN THE AUTHORS SUMMARY

Reviewer #2: At this stage, most remaining issues are editorial rather than substantive. The manuscript would benefit from careful copyediting to correct minor grammatical errors, duplicated punctuation, and a few awkwardly phrased sentences that interrupt the flow of reading. Consistency in terminology across the text and tables, especially for income and housing categories, would significantly improve readability without requiring additional analyses. The Data Availability Statement may also need slight refinement to more clearly indicate whether de identified data, a codebook, or metadata could be made available under controlled access, in line with journal expectations. These revisions are relatively minor but important for final clarity and compliance.

Reviewer #3: Accept

**Summary and General Comments**

Reviewer #1: (No Response)

Reviewer #2: Overall, this is a strong and thoughtful revision that clearly responds to previous reviewer feedback. The study addresses an important and underrepresented population, and the integration of epidemiological analysis, spatial data, and machine learning is a notable strength. The theoretical grounding in social determinants of health is clear, and the findings are both scientifically sound and socially relevant. The remaining weaknesses relate mainly to internal consistency, clarity of presentation, and minor methodological explanations rather than to the core design or validity of the study. With focused revisions addressing these points, the manuscript would be well positioned for acceptance.

Reviewer #3: The need to make it clear that the study does not focus on NTDs as listed by WHO should be clear enough to avoid confusion even if the added diseases have the same characteristic as NTDs.

PLOS authors have the option to publish the peer review history of their article (what does this mean? ). If published, this will include your full peer review and any attached files.

**Do you want your identity to be public for this peer review?** For information about this choice, including consent withdrawal, please see our Privacy Policy .

Reviewer #1: **Yes:** Melanie Awino Abongo

Reviewer #2: **Yes:** Dr.Arman Abdous DVM,MPH

Reviewer #3: No

**Figure resubmission:**

**Reproducibility:** To enhance the reproducibility of your results, we recommend that authors of applicable studies deposit laboratory protocols in protocols.io, where a protocol can be assigned its own identifier (DOI) such that it can be cited independently in the future. Additionally, PLOS ONE offers an option to publish peer-reviewed clinical study protocols. Read more information on sharing protocols at https://plos.org/protocols?utm_medium=editorial-email&utm_source=authorletters&utm_campaign=protocols

---

## [Editor Report · Decision Letter 2]

23 Feb 2026

Dear Ph.D Valentim,

We are pleased to inform you that your manuscript 'Social and environmental determinants of Neglected infectious diseases in quilombola communities of the brazilian amazon: an epidemiological and machine learning analysis' has been provisionally accepted for publication in PLOS Neglected Tropical Diseases.

Best regards,

Abdullahi Yusuf, PhD

Academic Editor

Elsio Wunder Jr

Section Editor

Shaden Kamhawi

co-Editor-in-Chief

Paul Brindley

co-Editor-in-Chief

---

## [Editor Report · Acceptance letter]

Dear Ph.D Valentim,

We are delighted to inform you that your manuscript, "Social and environmental determinants of Neglected infectious diseases in quilombola communities of the Brazilian Amazon: an epidemiological and machine learning analysis," has been formally accepted for publication in PLOS Neglected Tropical Diseases.

Best regards,

Shaden Kamhawi

co-Editor-in-Chief

Paul Brindley

co-Editor-in-Chief
